# Telework: Before and after COVID-19

**Mirela Cătălina Türkeş** [1],* and **Daniela Roxana Vuță** [2]

1   Faculty of Economics and Business Administration,
    Dimitrie Cantemir Christian University, 040051 Bucharest, Romania
2   Faculty of Economic Sciences and Business Administration,
    Transilvania University of Brasov, 1 Colina Universitatii Street, Building A, 500068 Brasov, Romania;
    daniela.vuta@unitbv.ro
*   Correspondence: mirela.turkes@ucdc.ro; Tel.: +40-728-176-475

**Definition:** Telework is, today, a voluntary form of work organization in which the employee is located outside the employer's premises, at home or elsewhere, under a telework contract, uses information and communication technologies (ICT) and works according to a predetermined schedule on the basis of an agreed supervisory mechanism and an online reporting system on the work undertaken.

**Keywords:** teleworking evolution; COVID-19; work legislation; work–life balance

## 1. History

During the COVID-19 pandemic, teleworking became the "new normal," changing people's lives and affecting working relationships. More than ever, it has inspired researchers from different fields of activity: economic, sociology, medical, ethical, etc. Reviewing the literature, this entry rediscovered the concept of "teleworking" and how it evolved in the context of socio-economic development from 2000 to 2019. In the context of the crisis generated by the COVID-19 pandemic, this entry looked for answers regarding the changes produced by teleworking, one of the main forms of adapting employment to social distancing. What changes have been made to the legal and contractual regulations? How was the work schedule organized and the working time recorded? How was the training provided and performance management carried out? What measures have been taken regarding the socio-fiscal protection, occupational safety, and work health of employees? What were the effects of digitalization? How has the work–life balance changed? In terms of results, there is clear evidence that telework leads to increased professional satisfaction, higher productivity, and reduced administrative costs, representing a viable future option.

Telework is a compound term originated by joining two components, "telou" (distance) from the Greek language and the word "tripalliare" (work) of Latin origin [1]. The term can also be found in the specialized literature under the appellation of remote work [2], working at a distance [3], teleworking [4], telecommuting [5], working from home (WFH) [6], mobile work [7], remote e-working [8], and work from anywhere [9]. It is worth noting that the term "telecommuting" appears almost exclusively in articles published in the North American media, while in European publications, the preferred term seems to be "telework."

The premises for the emergence of telework were created during the Industrial Revolution in telecommunications in the early nineteenth century. Once the federal postal service, telegraphs, photographs, and telephones were introduced, it generated the transition from a society based on voice communication, art, and writings (letters, newspapers, and books) to an industrialized one, marked by the development of the social division of labor, emergence of new branches of production, new industrial, commercial and transport centers, and expansion of economic, commercial, and communication relations.

Over time, researchers have tried to provide a unitary and well-accepted definition for telework, but with little success. A definition that covers the essence of telework and

differentiates it from other forms of work in the context of changes over time seems an impossible endeavor.

In the 1980s, Grant et al. (1985) [10] characterized teleworking as "one kind of remote working, or doing normal work activities while away from one's normal workplace." Other researchers have issued narrow definitions, such as: "working away from the central office" [11], "for employees to work out of their homes" [12], and "working at home" [13]. If computer and communications technology are considered [14], the definition of remote work is expanded to include aspects of the processes required to organize work outside the normal organizational boundaries of space and time.

In the second half of the twentieth century, the digital revolution, through the integration of digital computers and communication technologies, gradually allowed the transition from mechanical and analog electronic technology to digital electronics. The advent of the Internet, the introduction of the home computer, the invention of the World Wide Web, the provision of the first online services by its members, the transition to digital television, the use of mobile phones, online social networks, and the increase in the number of users have marked the beginning of an era of information and communication technologies in the production process, machines gradually diminishing the need for human intervention. In this context of the digital economy, the interest in teleworking has increased and international institutions, and contemporary researchers have developed new modern approaches to this concept.

In 1996, the International Labor Organization (ILO) saw work from home as a form of work organization in which the employee is located at a distance from the organization's headquarters or production units and cannot have personal contact with co-workers [15]. In 2020, the same organization expands the definition of telework, saying that it represents work performed outside the employer's headquarters, at the employee's home, or elsewhere by using information and communication technologies (ICT), such as smartphones, tablets, laptops, and desktop computers, etc., and carried out based on a voluntary agreement between the employer and the employee, based on some previously established hours/ schedule, agreed on surveillance mechanism, and some arrangements for reporting the work undertaken [16].

In addition to the above, in 2008, the European Commission proposed several recommendations on teleworking, such as voluntary choice of teleworking, the right to return to work at the organization's headquarters, the guarantee of maintaining the status of the employee, providing equal treatment, the employer's obligation to inform, train and evaluate the teleworker, coverage of the costs of teleworking arrangements by the employer, ensuring the protection of all teleworkers in the field of occupational safety and health (OSH), concluding insurance and confidentiality contracts with teleworkers, respecting the rights and obligations of teleworkers, and facilitating access to telework [17].

In the context of the European Employment Strategy, negotiated agreements between the European Council and several social partners for the modernization of the labor organization led to the issuance of Directives 91/533/EEC [18] and 2019/1152 [19], which regulated new teleworking-specific information, such as a telework employment agreement, telework voluntariness and reversibility, telework working instruments, equal treatment and non-discrimination of teleworkers, the privacy of teleworkers, and vulnerable groups of workers.

Some older studies present teleworking as a flexible way of working that involves the accomplishment of a wide range of remote work activities, electronic information processing, and using telecommunications to maintain the employer–employee relationship while performing full-time or part-time work [20,21]. Authors use terms such as "telework," "telecommuting," or "remote work," which generally refer to lucrative activities or work tasks carried out outside the office, either at home or elsewhere, and relying on new technologies [22–25].

The aim of the entry is to analyze the evolution of telework as a flexible and modern way of working before and after the COVID-19 pandemic, but also the psycho-socio-economic implications generated on the activity of European enterprises.

The first objective of the entry is to identify several specific aspects of telework, also highlighting its evolutionary change and socio-economic implications until 2019. Following

the change in the way we connect, communicate, and work, the second objective examines the changes in telework regarding legal and contractual regulations, work arrangements, working time, measures regarding socio-fiscal protection, work safety, the health of employees, digitalization, and work-life balance, all as a form of adapting to the sensitive socio-economic context generated by the COVID-19 pandemic. The third objective involves making a forecast of the evolution of the number of teleworkers between 2022 and 2025 in the EU (27 countries), which was made by using the Time Series Modeler procedure.

In the context of increasingly competitive and intensely digitized economies, this entry contributes to the literature by presenting the deepest transformations suffered by this voluntary form of work organization, from its appearance to the present.

### 2. The Evolution of Telework between 1990–2019

Current trends and future evolutions of telework have been the subject of numerous international studies. Specialists grouped the statistics related to telework, considering the degree of penetration in different countries, the potential for adaptation, and future growth. Thus, Huws (1991) [26] explored the meanings attributed to homeworkers and the present and future extent of teleworking. According to Gareis and Kordey (2000) [27], at the end of 1999, nine million EU residents were teleworking, and their number was to increase between 5–22.8% by 2005 in the most developed countries, such as France, Italy, Germany, Spain, and the United Kingdom.

From 2011 to 2019, the share of employed persons working from home in the total employment has experienced a different evolution from country to country. In Belgium, the share of employed persons working from home decreased from 9.9% in 2011 to 6.6% in 2018. In Figure 1, a similar evolution is observed in Denmark, where the share of employed persons working from home decreased from 12.0% (2011) to 7.9% (2019). At the opposite pole are Malta and Finland, where the share of employees working at home increased by 4.2 and, respectively, 3.6 in the same period [28].

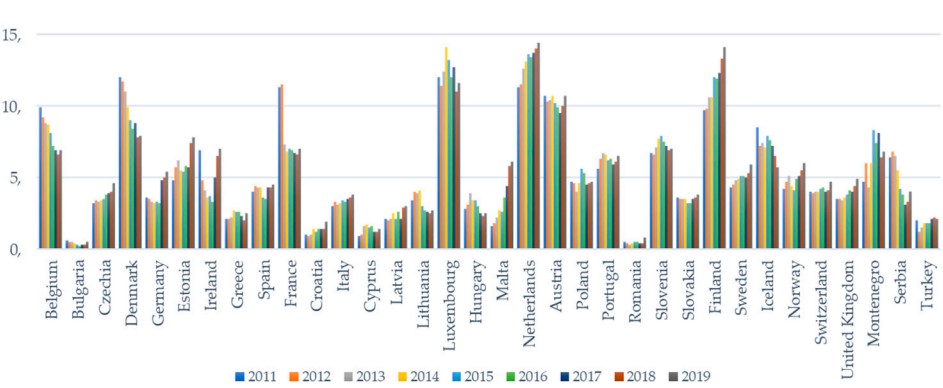

**Figure 1.** Employed persons working from home as a percentage of the total employment in 2011–2019.

In the European Union (EU), the share of people aged between 15 and 64 working from home out of the total employment increased from 5.4% in 2019 to 12.0% in 2020 and 13.2% in 2021 [28]. The highest shares of females working from home were recorded in Finland (13.3%), the Netherlands (12.5%), and Luxembourg (12.4%). A special situation is observed in the Netherlands (15.4%) and Finland (14.7%), where the share of males who worked from home during the pandemic was higher than that of females (see Figure 2).

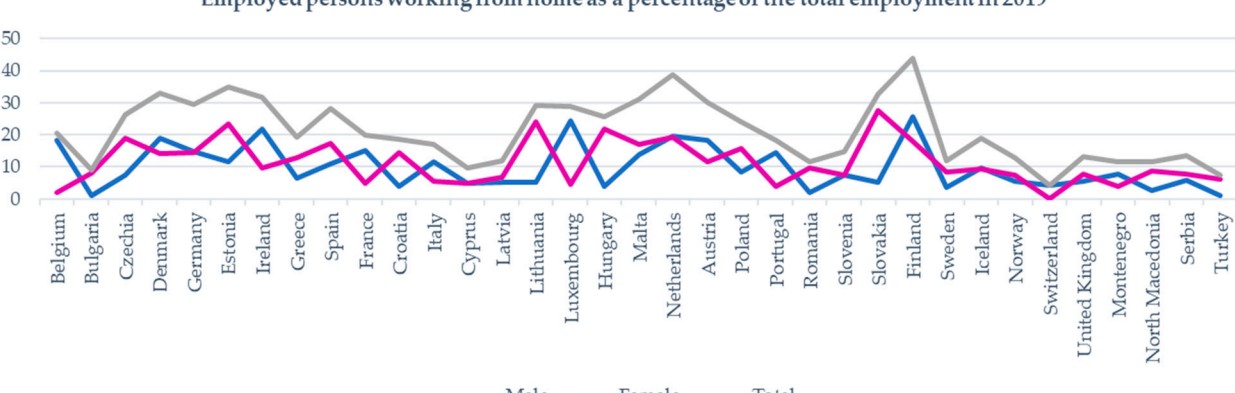

**Figure 2.** Employed persons working from home as a percentage of the total employment in 2019.

Dangelmaier et al., in 1999 [29], estimated that the penetration of teleworking in the U.S. would have increased to 25% by 2001. At the same time, other surveys indicated high rates of telework in Scandinavian countries and the Netherlands [30]. Grimes (2000) [31], analyzing the development perspectives of rural areas, considered that the social dimension must be connected with the technological dimension, focused on the installation of the necessary infrastructure and equipment, and that teleworking is the ideal form to be promoted in these areas. The results of a 1993 study showed that in New Zealand, the number of teleworkers had increased, especially among low-skilled technical workers, professionals, or innovative small businesses at home [32]. Another study from Latvia shows that, in the context of globalization, the integration of information and communication technologies, as well as the change of professional responsibilities, the demand for telework increases regardless of criteria, such as gender, age, and place of residence of the respondents. Moreover, the inhabitants of less populated areas are interested in socializing, and professional networking activities and are looking for a job in smart centers specially created for remote work [33]. Hesse (1995) [34] shows that the potential for teleworking will increase over time, presenting the example of the U.S. Department of Defense, which has expanded telecommuting arrangements for its employees with disabilities and designed the technologies needed for this form of work in the short and long term.

In recent years, more and more studies have shown that the use of teleworking brings many benefits to individuals, organizations, and society. These are presented in Table 1.

**Table 1.** The multiple benefits brought by telework.

| The Benefits Brought by Telework | |
| --- | --- |
| **1.** | **Individuals** |
| → contribute to achieving a better work–life balance;<br>→ obtaining a flexible schedule, working in a coworking space or on vacation;<br>→ decreasing the expenses regarding transport, clothes, and food;<br>→ reducing stress and burnout syndrome (professional burnout);<br>→ developing telework-specific skills. | [35–39] |
| **2.** | **Organizations** |
| → reducing administrative costs;<br>→ growing labour productivity;<br>→ increasing employee engagement;<br>→ finding new ways of employee recruitment. | [39–41] |

**Table 1.** *Cont.*

| The Benefits Brought by Telework | |
|---|---|
| 3. | **Society** |
| | → new job opportunities for rural residents and small communities; <br> → more time for parents to spend with their children while balancing their professional and personal interests; <br> → reduced commuting to and from work; <br> → facilitates its replacement with voluntary actions directed at people without food or shelter or with abandoned pets; <br> → reducing environmental pollution, consumption of non-renewable resources, and lowering waste levels;   [42] <br> → leads to reduced discrimination; <br> → increases the opportunity to find a job for people with health problems and veterans; <br> → improves public health; <br> → creates the opportunity to extend one's career beyond retirement. |

The existing literature highlights a series of studies that analyze the attitudes and perceptions of managers and employees related to telework. Some managers, although reporting positive attitudes towards telework, still show high resistance, although this could become a significant form of work [43]. Silva-C (2019) [44] noted that specific managerial practices, employee productivity, and the improvement of information security tools in organizations are the main factors affecting managers' attitudes towards adopting teleworking practices. For young workers who use information and communication technologies (ICT) easily, telework has become a natural part of work, while for older workers, it has become a form of postponing or staying active after retirement.

For enterprises, teleworking can also be seen as an opportunity to hire young and older people, ensure working patterns' diversification, and experience exchange [45].

**3. Telework and COVID-19 Pandemic**

With the outbreak of the COVID-19 pandemic, telework represented, on the one hand, a form of workforce adaptation to new social distancing conditions and, on the other hand, a method of organizing the activity of enterprises and employment. Telework rapidly extended, and many employees were forced to work from home, which, for a significant part of them, was a novelty. Moreover, the implementation of telework was one of the measures highly recommended or imposed by government institutions, which had to be accepted by employees and employers through agreements as a measure of maintaining an important part of the human capital and an opportunity to implement new informational communication technologies. In addition to the legal and contractual implications, both EU and national regulations on teleworking during the COVID-19 pandemic have undergone a series of changes related to work organization, tasks, working time, training, and performance management, and socio-fiscal protection, occupational safety and health measures, digitalization, and work–life balance.

*3.1. Legal and Contractual Implications*

In most EU countries, teleworking has been regulated by one of the following three methods: statutory legislation, social dialogue, and collective bargaining. If in some countries, there were statutory definitions and specific legislation included in the labor code and related rules (Belgium, Bulgaria, Czech Republic, Spain, Germany, Estonia, France, Greece, Hungary, Croatia, Italy, Lithuania, Luxembourg, Malta, the Netherlands, Poland, Portugal, Romania, Slovenia, and Slovakia), in others (Austria, Cyprus, Denmark, Finland, Ireland, Latvia, and Sweden), statutory legislation on teleworking has been regulated by laws on data protection, working time, and the safety and health of employees [46].

Individual and collective labor contracts were updated with new conditions in countries such as Romania, Bulgaria, Hungary, Luxembourg, and Slovenia, while in others (France, Estonia, Spain, Malta, Croatia, Italy, Portugal, and Greece) it was enough only to conclude written agreements. According to Hendrickx et al. (2020) [47], the Belgian government adopted a series of labor market-related measures, amending the legal regulations on teleworking and social distancing while implementing the Belgian temporary unemployment system. To reduce the effects of the pandemic, the French government introduced partial unemployment measures, thus distributing the efforts between the state, employers, and employees and limiting the loss of jobs [48].

### 3.2. Work Organization and Working Time

As the COVID-19 pandemic spread rapidly, more and more European enterprises were forced to introduce temporary leave, reduce wages, reduce working hours, temporarily or permanently suspend their activities, operate redundancies, or reduce the number of jobs. A significant part of the active European population was forced to choose between teleworking or part-time work, where the activities of the enterprises allowed it. In the case of teleworking, most of the changes included: work schedule modifications agreed upon by both parties or a pre-set schedule introduced by the employer, workload management, recording (monitoring) of the hours worked (limited, normal, and overtime), defining job autonomy with consequences on the priority and content of professional tasks, and others. Recent studies have shown that most teleworkers have experienced extended work schedules as a result of prolonged working hours during the evening and on weekends or due to the extension of meetings with clients or work teams to meet the demands of work [49–51]. Other theorists studying the effects of team member diversity on work dynamics and performance in the Global Virtual Team (GVT) showed that it generally has a substantial effect on effectiveness, while contextual diversity has a positive influence on task outcomes, and personal diversity has a negative impact on psychological outcomes [52–54]. Andrade and Petiz Lousã (2021) [55] showed that teleworking has brought about major changes in employees' work routines, such as increased overtime, overloaded roles, training on the use of technology after working hours, low work autonomy, and a blurred line between paid work and personal life.

### 3.3. Training and Performance Management

During the COVID-19 pandemic, enterprises identified new opportunities and faced new challenges related to staff training and professional improvement. On the one hand, telework required a different way of managing remote work teams and a distinct way of recruiting staff, while, on the other hand, it led to developing new skills needed to cope with a crisis, protecting the well-being of employees, and maintaining organizational performance.

Through the available devices and platforms, the enterprises had the opportunity to implement multiple initiatives, such as communication and collaboration with their employees, training programs to acquire the skills and knowledge necessary for successful teleworking, and motivation to support labor productivity and business performance. To foster performance, enterprises had to provide their employees with high-effective work equipment, adequate security conditions, training for leadership and communication skills in times of crisis, instructions for efficient time management, indications for minimizing health risks, options for maintaining the balance between work and personal life, and recommendations for adopting healthy eating habits and sufficient time for rest, etc. Buomprisco et al. (2021) [56] analyzed the effects of teleworking on workers' health and safety and highlighted the need to raise awareness among managers and teleworkers on the importance and right of having rest breaks during the working day, as well as the positive impact on health. Organizations had to teach their employees about the inherent risks in using data display screens [57], the use of e-learning platforms [58], or telementoring platforms for practical training [59].

One important challenge managers had to face was to maintain a high standard of performance, from the team level to the overall organizational performance, while fulfilling their commitments to employees, customers, stakeholders, and other beneficiaries. Kim et al. (2021) [60], assessing the role of supervisors in managing/ motivating teleworkers, showed that supervision, which includes results-based management and confidence-building efforts, has contributed substantially to improving organizational performance where teleworking has been applied. Buşu and György (2021) [61] demonstrated the influence of professional teleworking activities on employee performance in the context of adapting the new work system. Moreover, they claimed that the impact of teleworking on business performance was conditioned by several determining factors, such as the adequate management of human resources, capitalization of the positive aspects of teleworking, the ability to actively involve all staff, and the ability to reward teleworkers for the work performed and to prevent their resistance to change. The study conducted by Jamal et al. (2021) [62] showed that the presence of autonomy, the flexibility of working hours, and the existence of technological resources have contributed to improving the work–life balance, raising work productivity and performance, and increasing employees' satisfaction in full-time telework.

*3.4. Socio-Fiscal Protection, Occupational Safety, and Health*

With the transition to telework, employees have been exposed to various physical and psychological risks to their health and safety when working. For this reason, the organizations responsible for the health and safety of their teleworkers had to identify and effectively manage these occupational risks. During the COVID-19 pandemic, teleworking for most employees meant longer time spent in front of the computer.

Therefore, the physical health threats category includes the risks generated by prolonged office work in front of laptops. Ergonomic problems often lead to many diseases, such as scoliosis, myopia, diabetes, cardiovascular diseases, obesity, and/or can cause negative effects on the health of employees, such as back pain, neck pain, shoulder pain, tired eyes, etc. The psychological risks faced by teleworkers refer to isolation, overwork, stress, anxiety, depression, anger, insomnia, etc. [63].

Analyzing the significant impact of the intrusive type of management on employees during telework, Magnavita et al. (2021) [64] showed that performing tasks outside the working hours generated occupational stress, less happiness, anxiety, and depression. According to Brooks et al., 2021 [65], multi-week quarantine and social distancing had a strong impact on the entire population, leading to "emotional disorders, depression, stress, low mood, irritability, insomnia, post-traumatic stress symptoms, anger, and emotional exhaustion." Carillo et al., 2021 [49], highlighted how the crisis generated by the COVID-19 pandemic had profound implications on teleworkers, leading to professional isolation, a difficult work environment, and increased work and stress. Overall, the coronavirus pandemic has led to changes in occupational safety and health rules. Regarding how to promote employee health while working from home, several studies revealed the significant influence of teleworking on the mental and physical health of employees and proposed new recommendations for employers and employees to optimize their health (Oakman et al., 2020 and Turkes et al., 2021) [66,67]. Belzunegui-Eraso and Erro-Garcés (2020) [68] analyzed how to implement telework as a security practice to cope with the first wave of the COVID-19 disease crisis. Baert et al. (2020) [69] examined the perceptions of Flemish employees about teleworking and observed that some consider telework and digital conferencing as actions needed to retain employment during the COVID-19 crisis, while others admit that it diminishes their opportunities and weakens ties with their colleagues and their employer (43%).

Authorities have adopted socio-fiscal measures to help employees overcome the problems generated by the COVID-19 pandemic and to support those who carry out telework activities. For example, the Italian government targeted multiple measures, such as income support, parental leave, rest and/or leave, dismissal, and specific legislation to

maintain the health and safety of employees in telework but also to prevent the spread of the coronavirus at the workplace (Biasi*, 2020) [70].

According to Budacia (2021) [71], one of the social protection measures implemented in Romania during the pandemic was a non-taxable income within the limit of 400 lei per month, granted to employees who carried out teleworking activities to cover the expenses of utilities and purchases of office equipment. Cuerdo-Vilches (2021) [72] conducted an online survey to study the perception of Spanish employees about the workspace and its adequacy. The adequacy of teleworking spaces was insufficient for one-third of households, the number of people working or studying at home was extremely high, employees had to quickly improvise exclusive teleworking spaces, and the availability of digital resources was limited.

### 3.5. Digitalization

Digitalization, a socio-technical process that continuously evolves at the individual, organizational, societal, and global levels [73], is considered the key to innovation, competitiveness, and the growth of society [74]. Through its specific tools (cloud computing, software, platforms for teaching and learning, etc.) and complex technologies, digitalization creates the opportunity to increase organizational performance and work productivity, improve management practices, and create higher remuneration for new jobs created. In the context of teleworking and extensive usage of the Internet, the adoption of digital technologies by enterprises has led to new opportunities for teleworkers, a different way of organizing work, additional safety and health norms, new requirements for digital skills, labor standards, and a high interest towards the well-being of teleworkers. According to Stoica et al. (2021) [75], the spectacular development of information and communication technology corroborated with modern management (anthropocentric, systemic, product-oriented, etc.) led, in particular, to the creation of a new infrastructure for telework and, in general, to the creation of a global Internet network, in which digital technologies (cloud computing, Big Data, artificial intelligence, Internet of Things (IoT), Internet of Everything (IoE), etc.) are an integral part of the ecosystem of today's telemarketing paradigm. Sapfirova et al. 2021 [76], studying the impact of digitalization on legal regulations, labor relations, and the protection of labor rights, noticed the need to redefine the rights and obligations of employees and employers who have adopted this modern form of teleworking. It reiterated the employee's obligation to inform the employer about the equipment used during teleworking, as well as the employer's obligation to provide the teleworker with the equipment, technical support, and digital training necessary for carrying out the work activity.

### 3.6. Work–Life Balance

During the COVID-19 pandemic, telework was implemented in different circumstances than usual, more involuntary than voluntarily. For a significant part of the employees, work was carried out either remotely or working in the employer's location, full-time or part-time.

Few studies highlight the positive experiences employees had while working from home during the pandemic. Several employees admitted that telework has also produced positive effects, such as increased efficiency, a flexible work schedule [77], less commuting time, lower stress of travel to the employer's location, reduced risk of exhaustion [69], and new technology-related skills [78].

However, the biggest challenge for teleworkers during the pandemic remains the conflict between work and personal life. Recent studies show that maintaining a work–life balance has been an impossible task. Most employees said that teleworking generated a series of negative effects, such as an increased number of overtime hours, lack of hours reserved for rest and personal life, inefficient management of their workload [79], expansion availability of work-related functions, reduced well-being, higher risk of stress, etc. [56]. In part, these have created an inability to effectively disconnect from work due to a combination of daily stressors, which eventually leads to anxiety, burnout syndrome, or other associated illnesses [80].

Investigating the impact of telework on employee life and its contribution to the prospect of work–life conflict, Zhang et al. (2020) [81] demonstrated that children play a vital role in telework behavior as an important feature of the stage of family life that is complexly associated with telework behavior. Consequently, telework increases the conflict between "work–family" or "family–work," triggering the redistribution of domestic tasks in couples and aggravating gender differences. Before the pandemic, telework was associated with large gender differences, especially with housework and work interruptions for women. During the pandemic, more than a third of the employees worked from home, with the share of the female being higher than that of men. A recent UK study shows an illustrative case law relevant to the Equality Act, demonstrating that the challenges faced by employees, regardless of gender, are similar when teleworking is applied in the longer term [82].

The COVID-19 pandemic has profoundly affected the sphere of human life but also the protective behavior adopted by each country. Beyond the negative effects on mental health (anxiety, depression, stress, post-traumatic stress disorder, etc.), the crisis has affected the cultural dimension, creating more individualism, distance from power, uncertainty, and lack of transparency from public institutions [83–85].

An opinion poll among Austrian employees revealed that men have a more favorable attitude towards teleworking than females, and gender and age stereotypes have no effect on the 41% of Austrian employees who worked remotely during the pandemic [86–88]. At the end of 2020, the share of males (employees) working from home in the total employment in the EU reached 11.2%, while that of females reached 13%. (See Figure 3). In Romania, the share of employed persons working from home registered an upward trend, increasing from 0.8% in 2019 to 2.5% in 2020. The highest shares among the "male population" were recorded in Luxembourg (22.5%) and Ireland (21.3%), followed by the Netherlands (18.8%), Austria (17.5%), and Belgium (16.2%). On the opposite spectrum were situated countries such as Bulgaria (0.6%) and Turkey (0.9%). In terms of the "female population" who worked from home, the first places are occupied by Finland (25.5%) and Luxembourg (23.9%), and the last places are attributed to North Macedonia (3.8%), Romania (3.7%), and Bulgaria (1.9%) (See Figure 4) The share of men who worked from home continues to exceed that of the "females" in several countries, as follows: Netherlands (18.8% male; 16.7% female), Germany (13.8% male, 13.0% female), Iceland (9.1% male; 8.3% female) and Norway (5.1% male; 4.1% female).

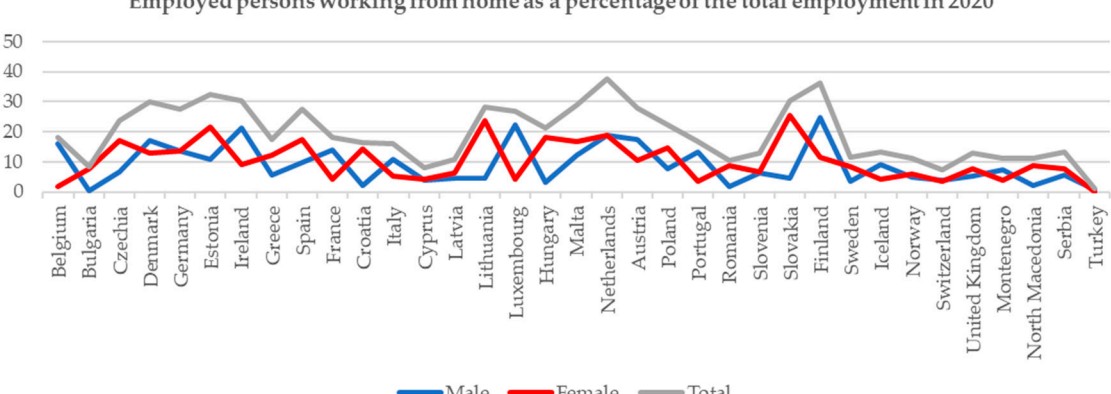

**Figure 3.** Employed persons working from home as a percentage of the total employment in 2020.

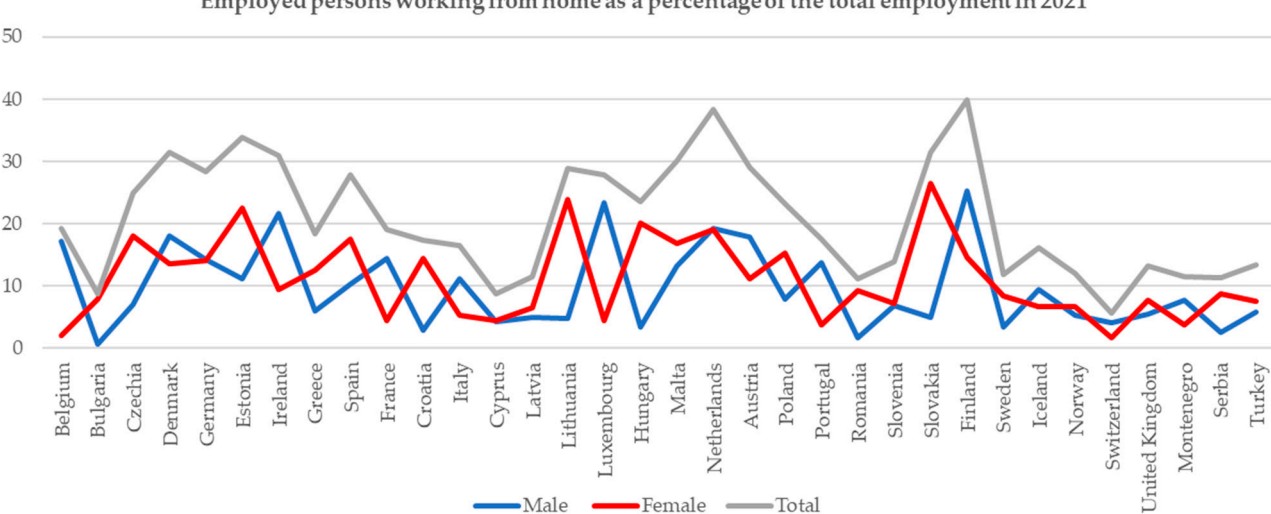

**Figure 4.** Employed persons working from home as a percentage of the total employment in 2021.

In 2021, the share of employees working from home in the total employment slowly continued its upward trend (see Figure 4), reaching maximum levels for men in Finland (25.8%), Luxembourg (24.3%), and Ireland (21.9%), while for women in the Netherlands (19.6%), Germany (14.7%), Iceland (11.2%) [28].

Given both the above-mentioned aspects and the database provided by Eurostat, a forecast has been made to highlight the future trends in the number of people who will work remotely (teleworkers) until 2025 in the EU (27 countries), using the Time Series Modeler procedure from the SPSS program. Figure 5 shows the clear, strong upward trend of the number of teleworkers in the EU from 2011 to 2021 compared to the previous period.

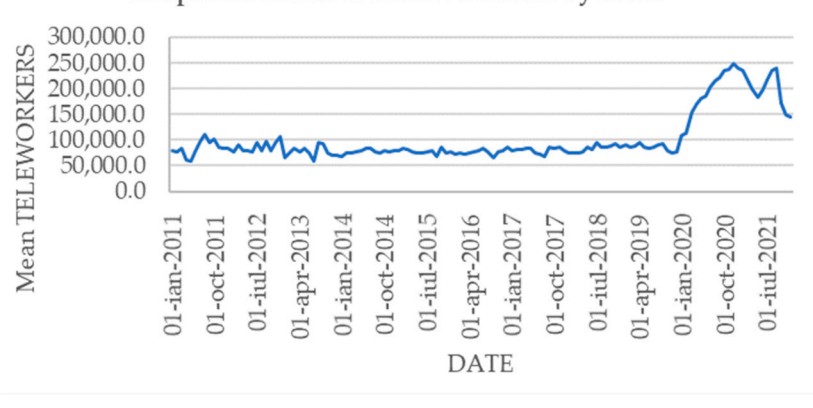

**Figure 5.** A simple line means of teleworkers in the UE during 2011–2021 by date.

In Table 2, the value of the stationary R-square (0.530 < 1) indicates a good, correct fit of the model and explains the variation observed in the Ljung–Box statistical series.

**Table 2.** Model fit statistics—teleworkers between 2022–2025.

| Model | Number of Predictors | Model Fit Statistics | Ljung–Box Q (18) | | | Number of Outliers |
|---|---|---|---|---|---|---|
| | | Stationary R-Squared | Statistics | DF | Sig. | |
| Teleworkers-Model_1 | 0 | 0.530 | 0.934 | 20.843 | 15 | 0.142 |

The significance value (0.142), greater than 0.05, is not significant; therefore, the model is specified correctly.

Figure 6 indicates the observed values of the dependent series, the values for the forecast period, the values for the estimation period, and the confidence intervals for the forecast period. In conclusion, the model predicted an upward evolution of the number of teleworkers in the EU for the forecasted period from 2022–2025.

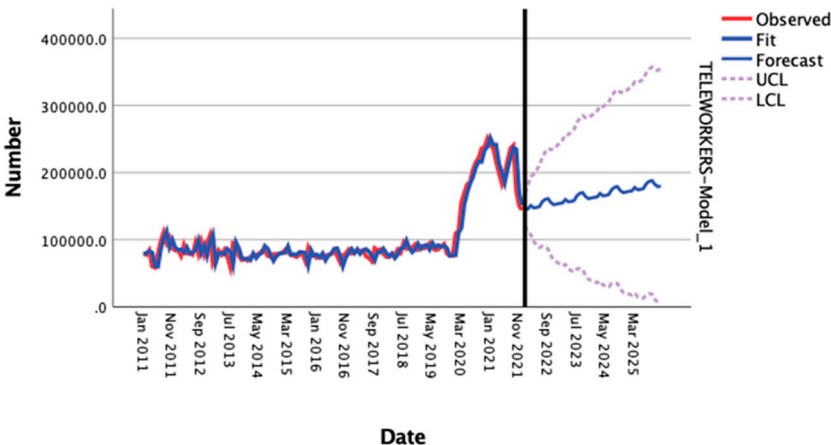

**Figure 6.** Predicted and observed values—teleworkers between 2011–2025.

## 4. Future Direction

While before the COVID-19 pandemic, telework seemed an unrealizable dream for some employees, today it represents the future of work. Rejected in the past, due to the lack of certainties regarding the implementation of labor relations and due to a low degree of technology, and temporarily recommended by governments during the state of alert and imposed unilaterally on employees, telework required a complete, modern design for the professional activity to be carried out at optimal parameters. Moreover, it came as a vital alternative for enterprises to ensure business continuity and performance standards.

By 2020, many enterprises have implemented telework in full or temporarily or have introduced mixed working solutions, in which a physical presence at the office alternates with telework, and most likely, they intend to use this viable model in the future. Regardless of the official recommendations for physical distance or the limitation of social interactions, implementing telework within enterprises was mainly possible due to five recognizable advantages of teleworking, namely: tested productivity, employee motivation, reduced employee fluctuation, reduction of administrative costs, and environmental protection.

Given that more and more employees are working from home, teleworking is no longer a Millennial trend but a way of life, where employees are happier and more productive, and organizations operate more efficiently and profitably.

Further research should focus, during and after the COVID-19 crisis, both on the evaluation of the situation of remote work within organizations and on the knowledge of the perceptions of Romanian employees regarding the use of telework.

**Author Contributions:** Conceptualization, M.C.T.; methodology, M.C.T.; software, M.C.T. and D.R.V.; validation, M.C.T.; formal analysis, M.C.T.; investigation, M.C.T. and D.R.V.; resources, M.C.T. and D.R.V.; data curation, M.C.T.; writing—original draft preparation, M.C.T. and D.R.V.; writing—review and editing. All authors have read and agreed to the published version of the manuscript.

**Funding:** This research received no external funding.

**Institutional Review Board Statement:** Not applicable.

**Informed Consent Statement:** Not applicable.

**Data Availability Statement:** Not applicable.

**Conflicts of Interest:** The author declares no conflict of interest.

**Entry Link on the Encyclopedia Platform:** https://encyclopedia.pub/entry/25214.

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
