# Peer review of "Telework: Before and after COVID-19"

_encyclopedia, doi:10.3390/encyclopedia2030092_

Round 1
Reviewer 1 Report
- The keywords of the article are recommended to select and delete someone.
- Figures 2, 3, and 5 seem not very clear, and it is recommended to redraw; in addition, the legend of the figure is recommended to be added.
- The benefits of teleworking on page 5 are suggested and presented in a table that is easier for Encyclopedia readers to understand.
- This article comprehensively discusses the epidemic's impact and analyzes changes in many aspects. The pandemic has changed many things, making teleworking work a daily norm. The work of correlation analysis can also be used as a reference for follow-up research.
Author Response
Review Report (Round 1)
- The keywords of the article are recommended to select and delete someone.
Response: The necessary changes have been made, according to the reviewer's recommendations.
- Figures 2, 3, and 5 seem not very clear, and it is recommended to redraw; in addition, the legend of the figure is recommended to be added.
Response: Figures 2, 3, and 5 have been reshaped to provide more clarity on the information provided.
- The benefits of teleworking on page 5 are suggested and presented in a table that is easier for Encyclopedia readers to understand.
Response: The benefits of working remotely (page 5) have been presented in a table, as the reviewer suggested. This would ensure more clarity and easiness in understanding for the Encyclopedia readers.
- This article comprehensively discusses the epidemic's impact and analyzes changes in many aspects. The pandemic has changed many things, making teleworking work a daily norm. The work of correlation analysis can also be used as a reference for follow-up research.
Response: As the reviewer suggested, the following paragraph was inserted (lines 436-438):
„Further research should focus, during and after the COVID-19 crisis, both on the evaluation of the situation of remote work within the organizations, and on the knowledge of the perceptions of the Romanian employees regarding the use of telework”.

Reviewer 2 Report
- The format of the whole manuscript, especially the Reference, does not conform to the format of Encyclopedia. The authors should revise the whole paper according to the instructions and the template of the journal carefully. Moreover, there should be more references from Encyclopedia.
- The objectives of the study should be clearly stated in Abstract and Introduction.
- In Introduction, authors should cite the latest literature to explain the topic of this research [1-2].
- The unique innovations and contributions of this paper should be further emphasized.
- If there are similar studies, a comparison of the results would be indicated.
Author Response
Review Report (Round 2)
- The format of the whole manuscript, especially the Reference, does not conform to the format of Encyclopedia. The authors should revise the whole paper according to the instructions and the template of the journal carefully. Moreover, there should be more references from Encyclopedia.
Response:
1a. Both the format of the manuscript and references have been revised.
1b. Three new references from Encyclopedia were included and the following text has been added:
„The COVID-19 pandemic has profoundly affected the sphere of human life but also the protective behavior adopted by each country. Beyond the negative effects on mental health (anxiety, depression, stress, and post-traumatic stress disorder, etc.), the crisis has affected the cultural dimension, creating more individualism, distance from power, uncertainty, and lack of transparency from public institutions. [83-85]” (Lines 373-377)
[83] Balcombe, L.; De Leo, D. Digital Mental Health Amid COVID-19. Encyclopedia 2021, 1, 1047-1057. https://doi.org/10.3390/encyclopedia1040080
[84] Bueno-Guerra, N. COVID-19 and Psychological Impact. Encyclopedia 2022, 2, 400-408. https://doi.org/10.3390/encyclopedia2010024
[85] Nair, N.; Selvaraj, P.; Nambudiri, R. Culture and COVID-19: Impact of Cross-Cultural Dimensions on Behavioral Responses. Encyclopedia 2022, 2, 1210-1224. https://doi.org/10.3390/encyclopedia2030081
- The objectives of the study should be clearly stated in Abstract and Introduction.
Response: Based on the recommendation of the reviewer, the following paragraph was included in the "Introduction" section. Being an entry, the abstract was removed.
„The aim of the paper is to analyze the evolution of telework - as a flexible and modern way of working, before and after the COVID-19 pandemic, but also of the psycho-socio-economic implications generated on the activity of European enterprises.
The first objective of the paper is to identify several specific aspects of telework highlighting also its evolutionary change and socio-economic implications until 2019. Following the change in the way we connect, communicate and work, the second objective examines the changes in telework regarding legal and contractual regulations, work arrangements and working time, measures regarding socio-fiscal protection, work safety, and health of employees, digitalization and work-life balance, all as a form of adapting to the sensitive socio-economic context generated by the COVID-19 pandemic. The third objective involves making a forecast of the evolution of the number of teleworkers between 2022 and 2025 in the EU (27 countries) was made, by using the Time Series Modeler procedure.”
- In Introduction, authors should cite the latest literature to explain the topic of this research [1-2].
Response: At the request of the reviewer, the sources below have been included:
[87] Yang, W.; Gao, H.; Yang, Y.; Liao, J. Embodied Carbon in China’s Export Trade: A Multi Region Input-Output Analysis. Int. J.Environ. Res. Public Health 2022, 19, 3894. https://doi.org/10.3390/ijerph19073894
[88] Yang, W.; Gao, H.; Yang, Y. Analysis of Influencing Factors of Embodied Carbon in China’s Export Trade in the Background of “Carbon Peak” and “Carbon Neutrality”. Sustainability 2022, 14, 3308. https://doi.org/10.3390/su14063308
- The unique innovations and contributions of this paper should be further emphasized.
Response: As the reviewer suggested, the following paragraph has been inserted in the "Introduction" section:
„In the context of increasingly competitive and intensely digitized economies, this paper contributes to the literature by presenting the deepest transformations suffered by this voluntary form of work organization, from its appearance to the present.”
- If there are similar studies, a comparison of the results would be indicated.
Response: The authors consider that there are no similar studies, therefore no comparison of the results can be made.
Reviewer 3 Report
The article addresses issues related to telework in the context of the COVID-19 pandemic.
There are some minor editorial errors.
The History paragraph should be numbered 1.
Captions of figures end with dot.
There is no reference in the text for Figure 1 and Figure 5
In Figures 3, 4, 5 is missing (c).
If the figures are not original the source should be indicated.
Instead [39, 40-41] you may use [39-41].
Paragrapf 2 Covid should be COVID as it is abbreviation.
Line 383 " female population" there is a blank
Lines 520, 539 you should delete pp.
DOI sould be used on similar format in each reference.
Otherwise, the article corresponds to the publication rigors.
Author Response
Review Report (Round 3)
- The article addresses issues related to telework in the context of the COVID-19 pandemic.
- There are some minor editorial errors.
- The History paragraph should be numbered 1.
Response: All sections of the paper have been re-numbered.
- Captions of figures end with dot.
Response: The requested changes have been made.
- There is no reference in the text for Figure 1 and Figure 5
Response: Text references have been introduced for Figure 1 and Figure 5.
- In Figures 3, 4, 5 is missing (c).
Response: The requested changes have been made.
- If the figures are not original the source should be indicated.
Response: All figures have been redone accordingly.
- Instead [39, 40-41] you may use [39-41].
Response: The requested changes have been made.
- Paragrapf 2 Covid should be COVID as it is abbreviation.
Response: COVID abbreviation was considered and included.
- Line 383 " female population" there is a blank
Response: The free row was eliminated.
- Lines 520, 539 you should delete pp.
Response: „pp.” was deleted.
- DOI sould be used on similar format in each reference.
Response: DOI was integrated with the format as found and published online (links) for most bibliographic sources. Older works do not include DOI.
- Otherwise, the article corresponds to the publication rigors.
We are most thankful!
Reviewer 4 Report
Thank you for the opportunity to revise the paper titled: “Telework: before and after COVID-19”.
First if all, I would like to congratulate the authors for a well written article, dealing with a topic that is clearly relevant nowadays. I have some comments that I hope the authors find constructive:
The paper covers interesting literature about telework, but I think perhaps it could be complemented with the existing literature on global virtual teams (GVTs). These literature, although combining also aspects related to multiculturality, has analyzed the increasing trend of remote working, as well as its advantages and challenges (see, for instance, Taras et al. 2013 and 2019, and Jiménez et al. 2017).
The paper focuses often on the European Union, both in the text and the maps/figures. Would it be possible to include more data on other areas? If not, perhaps this focus on the European context should be mentioned in the abstract of the paper, so readers can know what to expect.
The last section, about future directions, seems too short and I think expanding it would increase the value of the article.
As a minor thing, I would recommend numbering the first section as 1), and the rest accordingly.
References:
Jimenez, A., Boehe, D. M., Taras, V., & Caprar, D. V. (2017). Working across boundaries: Current and future perspectives on global virtual teams. Journal of International Management, 23(4), 341-349.
Taras, V., Caprar, D. V., Rottig, D., Sarala, R. M., Zakaria, N., Zhao, F., ... & Huang, V. Z. (2013). A global classroom? Evaluating the effectiveness of global virtual collaboration as a teaching tool in management education. Academy of Management Learning & Education, 12(3), 414-435.
Taras, V., Baack, D., Caprar, D., Dow, D., Froese, F., Jimenez, A., & Magnusson, P. (2019). Diverse effects of diversity: Disaggregating effects of diversity in global virtual teams. Journal of International Management, 25(4), 100689.
Author Response
Review Report (Round 4)
- Thank you for the opportunity to revise the paper titled: “Telework: before and after COVID-19”.
- First if all, I would like to congratulate the authors for a well written article, dealing with a topic that is clearly relevant nowadays. I have some comments that I hope the authors find constructive:
- The paper covers interesting literature about telework, but I think perhaps it could be complemented with the existing literature on global virtual teams (GVTs). These literature, although combining also aspects related to multiculturality, has analyzed the increasing trend of remote working, as well as its advantages and challenges (see, for instance, Taras et al. 2013 and 2019, and Jiménez et al. 2017).
Response: At the request of the reviewer, the following paragraph and sources were included:
Taras et al. (2019) studying the effects of team member diversity on work dynamics and performance the Global Virtual Team (GVT) showed that it generally has a substantial effect on effectiveness, while contextual diversity has a positive influence on task outcomes and personal diversity has a negative impact on psychological outcomes.
[52] Jimenez, A.; Boehe, D.M.; Taras, V.; Caprar, D.V. Working across boundaries: Current and future perspectives on global virtual teams. Journal of International Management, 2007, 23(4), 341-349. https://doi.org/10.1016/j.intman.2017.05.001
[53] Taras, V.; Caprar, D.V.; Rottig, D.; Sarala, R.M.; Zakaria, N.; Zhao, F.; Jiménez, A.; Wankel, C.; Lei, W.S.; Minor, M.S.; BryÅ‚a, P. A global classroom? Evaluating the effectiveness of global virtual collaboration as a teaching tool in management education. Academy of Management Learning & Education, 2013, 12(3), 414-435. https://doi.org/10.5465/amle.2012.0195
[54] Taras, V.; Baack, D.; Caprar, D.; Dow, D.; Froese, F.; Jimenez, A.; Magnusson, P. Diverse effects of diversity: Disaggregating effects of diversity in global virtual teams. Journal of International Management, 2019, 25(4), 100689. https://doi.org/10.1016/j.intman.2019.100689
- The paper focuses often on the European Union, both in the text and the maps/figures. Would it be possible to include more data on other areas? If not, perhaps this focus on the European context should be mentioned in the abstract of the paper, so readers can know what to expect.
Response: Consequently, the following content was introduced:
„The aim of the paper is to analyze the evolution of telework - as a flexible and modern way of working, before and after the COVID-19 pandemic, but also of the psycho-socio-economic implications generated on the activity of European enterprises”.
- The last section, about future directions, seems too short and I think expanding it would increase the value of the article.
Response: Consequently, the following text below has been introduced:
„Further research should focus, during and after the COVID-19 crisis, both on the evaluation of the situation of remote work within the organizations, and on the knowledge of the perceptions of the Romanian employees regarding the use of telework”.
- As a minor thing, I would recommend numbering the first section as 1), and the rest accordingly.
Response: All sections of the paper have been renumbered.
Reviewer 5 Report
I want to thank the authors for this interesting submission on the current status of telework and how its status has changed due to the pandemic. Overall the paper is well written, but there are still some issues that should be addressed before it is ready for publication.
On p. 2, the authors argue that telework is not only work from home, yet they later on then only focus on work from home. At this point the authors should present their working definition for telework that is used throughout the article to make sure that their view on the topic is clear and remains consistent.
Next, the authors should at least briefly address how they came up with the topics that are highlighted in section 2. Was this the result of an actual systematic review or just some narrative summary of some random articles. It is not clear what system was used here and how the authors came up with it.
Regarding the applied methodology, the time series analysis used on p. 10 and onward also needs more details. It should be explained what is shown in Table 1 and Figure 7 and has to be explained why this analysis is needed as part of this study. If the authors decide to keep the analysis and its description in its current state, I would rather suggest that it is removed from the paper in favor of adding further details on the methodology that was used to extract the topics highlighted in section 2.
There are also some references missing in the paper, with some examples including (but not limited to):
- 7 line 279 onward: Reference missing for diseases caused by ergonomic problems
- 10 line 400 onward: Reference for Time Series Modeler procedure missing
Then, the authors should also revise their Figures to be suitable for publication.
Figure 1 in general provides an almost incomprehensible overview of the development of telework in the EU. In addition, it is even misleading, as gaps in data are just filled up (see North Macedonia). I would suggest that the authors use a line graph or something similar to show the overall development or perhaps only pick some representative countries and an overall average for all countries to give an overview of the development.
Figure 2 (and others): resolution not sufficient (legend is not readable)
Figure 3: decimal symbol not consistent (comma instead of dot)
Figure 4 a, b + 5 a, b: numbers in some cases not visible
Figure 6: authors claim that a clear, strong upward trend is shown in the period 2011 to 2021. We do not see the previous period and there is no clear upward trend for 2011-2021. Rather, the share is stagnating with the exception of the pandemic years. In addition, the scale for the y-axis is likely in Millions, which should be stated as such.
Finally, some minor proofreading is needed. For example (problematic areas in brackets):
In abstract: “…changes [suffered] from teleworking…”
In “Future direction”: “Rejected in the past, due to the [lack of uncertainties]…”
Overall a highly relevant topic, though there is still some effort needed to create a good overview on how telework has changed due to the pandemic.
Author Response
Review Report (Round 5)
I want to thank the authors for this interesting submission on the current status of telework and how its status has changed due to the pandemic. Overall the paper is well written, but there are still some issues that should be addressed before it is ready for publication.
On p. 2, the authors argue that telework is not only work from home, yet they later on then only focus on work from home. At this point the authors should present their working definition for telework that is used throughout the article to make sure that their view on the topic is clear and remains consistent.
Response: Considering the reviewer's recommendation, a definition of "telework" was introduced.
„Definition: Telework is today a voluntary form of work organization in which the employee is located outside the employer's premises, at home or elsewhere, under a telework contract, uses information and communication technologies (ICT), works according to a predetermined schedule, on the basis of an agreed supervisory mechanism and an online reporting system on the work undertaken”.
Next, the authors should at least briefly address how they came up with the topics that are highlighted in section 2. Was this the result of an actual systematic review or just some narrative summary of some random articles. It is not clear what system was used here and how the authors came up with it.
Regarding the applied methodology, the time series analysis used on p. 10 and onward also needs more details. It should be explained what is shown in Table 1 and Figure 7 and has to be explained why this analysis is needed as part of this study. If the authors decide to keep the analysis and its description in its current state, I would rather suggest that it is removed from the paper in favor of adding further details on the methodology that was used to extract the topics highlighted in section 2.
Response: It is the result of a real systematic review, using the classic system.
There are also some references missing in the paper, with some examples including (but not limited to):
- 7 line 279 onward: Reference missing for diseases caused by ergonomic problems
Response: The bibliographic source below has been introduced.
„[63] de Macêdo, T.A.M.; Cabral, E.L.D.S.; Silva Castro, W.R.; de Souza Junior, C.C.; da Costa Junior, J.F.; Pedrosa, F.M.; da Silva, A.B.; de Medeiros, V.R.F.; de Souza, R.P.; Cabral, M.A.L.; Másculo, F.S. Ergonomics and telework: A systematic review. Work, 2020, 66(4), 777-788, DOI: 10.3233/WOR-203224”
- 10 line 400 onward: Reference for Time Series Modeler procedure missing
Response: Time Series Modeler procedure is an analysis performed within the SPSS program, and does not require a reference.
Then, the authors should also revise their Figures to be suitable for publication.
Figure 1 in general provides an almost incomprehensible overview of the development of telework in the EU. In addition, it is even misleading, as gaps in data are just filled up (see North Macedonia). I would suggest that the authors use a line graph or something similar to show the overall development or perhaps only pick some representative countries and an overall average for all countries to give an overview of the development.
Figure 2 (and others): resolution not sufficient (legend is not readable)
Figure 3: decimal symbol not consistent (comma instead of dot)
Figure 4 a, b + 5 a, b: numbers in some cases not visible
Figure 6: authors claim that a clear, strong upward trend is shown in the period 2011 to 2021. We do not see the previous period and there is no clear upward trend for 2011-2021. Rather, the share is stagnating with the exception of the pandemic years. In addition, the scale for the y-axis is likely in Millions, which should be stated as such.
Response: All figures have been redone and renumbered.
Finally, some minor proofreading is needed. For example (problematic areas in brackets):
In abstract: “…changes [suffered] from teleworking…”
Response: The word "suffered" has been replaced with "produced".
In “Future direction”: “Rejected in the past, due to the [lack of uncertainties]…”
Response: “Rejected in the past, due to the lack of certainties…”
Overall a highly relevant topic, though there is still some effort needed to create a good overview on how telework has changed due to the pandemic.
Thank you.
Round 2
Reviewer 1 Report
The authors made a positive response to all comments. Therefore, I have no further comment.
Reviewer 2 Report
Authors have addressed the comments in the revised version of manuscript. The current version is appropriate for publication.
Reviewer 3 Report
There are some minor editing errors that need correction.
Reviewer 4 Report
No further comments
Reviewer 5 Report
I want to thank the authors for their revision, which has sufficiently addressed the issues I have raised.
The only thing I can still recommend is that the authors should spend more time on their response letter. On face value, I expected that some of my comments were ignored, but were then actually addressed in the revision. This may lead to unnecessary misunderstandings in the future.
Congratulations on the publication and all the best for your future research!